# Enhanced Performance of Immobilized *Rhizopus oryzae* Lipase on Coated Porous Polypropylene Support with Additives

Fatimah Sani [1], Nur Fathiah Mokhtar [1], Mohd Shukuri Mohamad Ali [1,2] and Raja Noor Zaliha Raja Abd Rahman [1,2,3,4,*]

1 Enzyme and Microbial Technology Research Centre (EMTech), Faculty of Biotechnology and Biomolecular Science, Universiti Putra Malaysia, Serdang 43400, Selangor, Malaysia; fatimahsani@yahoo.com (F.S.); nurfathiah93@gmail.com (N.F.M.); mshukuri@upm.edu.my (M.S.M.A.)
2 Department of Biochemistry, Faculty of Biotechnology and Biomolecular Science, Universiti Putra Malaysia, Serdang 43400, Selangor, Malaysia
3 Department of Microbiology, Faculty of Biotechnology and Biomolecular Science, Universiti Putra Malaysia, Serdang 43400, Selangor, Malaysia
4 Institute of Bioscience, Universiti Putra Malaysia, Serdang 43400, Selangor, Malaysia
* Correspondence: rnzaliha@upm.edu.my

**Abstract:** The immobilization of *Rhizopus oryzae* lipase (RoL) by hydrophobic adsorption on polypropylene supports with additives was investigated. Additives such as hen egg albumin, sodium caseinate and CAVAMAX® W6 were used to coat the support during immobilization where the immobilized RoL on coated support was compared to those of noncoated support. Following the immobilization, the catalytic activity of immobilized RoL was characterized based on different temperatures and pH. The immobilized RoL without additives showed optimal lipase activity at an optimum temperature of 50 °C and pH 6. However, RoL lipase that was immobilized on support treated with CAVAMAX® W6 had better performance in terms of hydrolytic activity and stability as compared to other additives. In addition, by having a support treated with hen egg albumin, the immobilized RoL was capable of yielding higher ester during esterification reactions.

**Keywords:** polypropylene; additive; *Rhizopus oryzae*; lipase

## 1. Introduction

Lipases (triacylglycerol hydrolases, EC 3.1.1.3) belong to the enzyme class of hydrolases catalyzing the hydrolysis of the triglycerides at the lipid water interface into fatty acids, mono- and diacylglycerols, and glycerol. In terms of application, lipases have been widely used in the food industry and have a highly selective reaction towards a wide range of substrates. Lipases with 1,3-positional selectivity play an essential role in synthesizing 1,3-diacylglycerol and structured lipids [1]. However, said lipases are highly unstable in an aqueous form, which requires immobilization to form immobilized lipase with enhanced activity, thermal, reusability, and operational stability. Various immobilization methods have been applied to lipase, such as adsorption, entrapment, covalent binding, and crosslinking in which the most applied method for lipase immobilization is physical adsorption. However, the interaction between support and lipase via physical adsorption is weak and prone to leaching out from the supports [2,3].

The immobilization of lipases on engineered supports could refrain the free movement of the lid or allow the enzyme to immobilize in an open form where the immobilized lipase remains active. The activation of *Rhizopus oryzae* lipase occurred when the condition of immobilization includes the hydrophobic surface of the support that displaces the lid, permitting the open conformation of lipase [4]. As the support becomes highly hydrophobic, the inner pores of the support turned out to be inaccessible to the aqueous enzyme solution. Most microbial enzymes have different pore sizes that ranged from 4 to 5 nm. Hence,

the size of the pore and porosity would contribute toward localization of enzyme on the support [5].

Polypropylene support is well known for adsorption of lipase and has been extensively utilized for the immobilization. For example, a locally purchased MP1008 is a porous polypropylene commercial support and had been used for immobilizing several lipases derived from *Candida antarctica* [6], *Burkholderia cepacian* [7], *Mucor javanicus* [8], *Candida rugosa* [9], *Aspergillus niger*, *Mucor javanicus* and *Penicillium roqueforti* (A, M and R lipase) [10]. In general, the support provides good immobilization performance due to its irregular shape, high hydrophobicity and porosity. As for applications, the support benefited industry in terms of a cost-saving approach that links to the preparation of biocatalyst and new applications [6]. However, there is also limitation on the utilization of readily available supports as the existing one contributes to the loss of hydrolytic activity of the immobilized *Rhizopus oryzae* lipase (rROL) [11].

Therefore, by coating support with additives, the interactions of the enzyme with the support can be improved and simultaneously enhance the lipase activity and stability. In addition, further loss of enzyme activity after immobilization can be avoided by adding additives during immobilization, affecting the microenvironment of the enzyme. According to [12], β-cyclodextrin has shown positive results in the biocatalytic activity of *Yarrowia lipolytica* Lipase 2 (YLLIP2) in organic solvents [12]. Undesirable interactions of similar-size or larger proteins with the support surface can be prevented using a medium-sized protein, bovine serum albumin, following the disclosure by Bolivar et al. [13], which eases the adsorption of smaller proteins on the support.

The use of immobilized lipases in non-aqueous reaction media is highly preferred over aqueous solutions for industrial application where a reverse reaction can be applied for biocatalysis. In the present study, the immobilized RoL was subjected to esterification on oleic acid to produce ethyl oleate. Ethyl oleate is a fatty ester commonly used as a fuel additive and as a solvent for pharmaceutical drug preparation. Therefore, this study aimed to investigate the effect of the additive coatings on the polypropylene support for the purpose of immobilizing *Rhizopus oryzae* lipase, where the difference of immobilized lipase on different additive supports was determined based on its activity and ability to perform esterification with high ester conversion.

## 2. Results

### 2.1. Immobilization of RoL in Polypropylene Support

The immobilization of RoL via adsorption technique in the presence of additives for surface coating polymers was studied. The additives used were hen egg albumin, sodium caseinate and CAVAMAX® W6, which were of food grade. Previously, hen egg albumin was used as an additive in the immobilization of RoL according to Bosley and Peilow, 1993 [14]. The effect of adding additives on the RoL immobilization on a polypropylene support was shown in Figure 1. Based on Figure 1, about 96% of RoL lipase was adsorbed in the support treated with hen egg albumin, followed by 89% with CAVAMAX® W6 and 71% with sodium caseinate. As for untreated support, it was less than 10% of lipase being physically adsorbed.

Although hen egg albumin contributed towards high yield of RoL immobilization, a significant loss of 91% was observed in the hydrolytic activity of RoL after immobilization on the same supports with same additive (Table 1). As for supports treated with sodium caseinate and CAVAMAX® W6, 43% and 51% of enzymes remained adsorbed to respective supports. It shows that the strong binding interaction between enzyme and the support via multiple attachments was obtained.

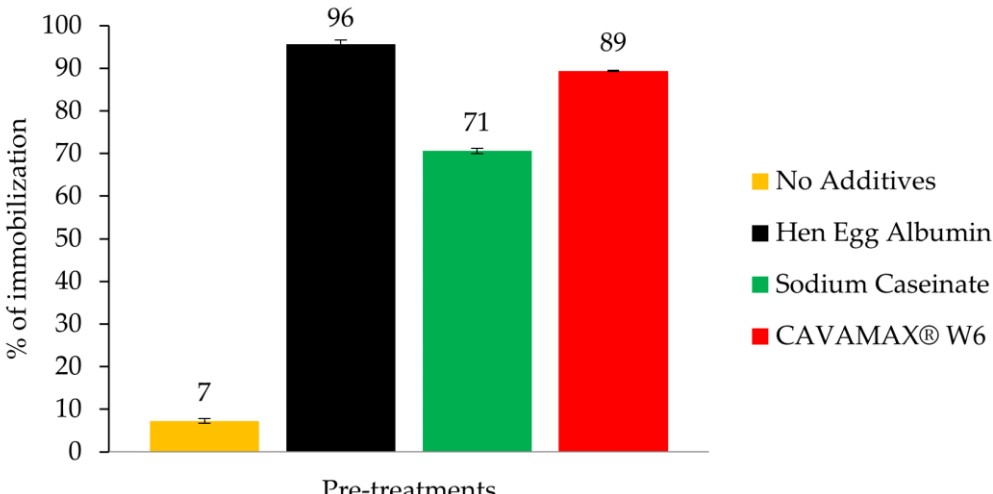

**Figure 1.** Immobilization yield of *Rhizopus oryzae* lipase (RoL) immobilized on no additives and additive-treated polypropylene support. Additives use in this study were hen egg albumin, sodium caseinate and CAVAMAX® W6. The measurement of yield of immobilization was conducted in triplicate and relative activities are measured as mean values ± standard error.

**Table 1.** The percentage of enzyme leaching from immobilized RoL.

| Supports/Treatments | Percentage of Loss after Immobilization (%) |
| --- | --- |
| No Additives | 0 |
| Hen egg albumin | 91 |
| Sodium caseinate | 43 |
| CAVAMAX® W6 | 51 |

*2.2. Biocatalytic Properties of Immobilized RoL*

2.2.1. Effect of Temperature on Lipase Activity

According to Figure 2, the relative activity of free and immobilized RoL with the effect of various assay temperatures was determined and compared. The activity of RoL immobilized on support treated with hen egg albumin achieved the lowest activity at 20 to 30 °C, while at 40 to 50 °C, the relative activity of RoL increased up to 80% and 100%, respectively. The immobilized lipase on CAVAMAX® W6 support attained the highest lipase activity followed by sodium caseinate at 435.7 and 311.8 U/g, respectively. However, as the temperature increased to 60 °C, lipase activity of immobilized RoL with CAVAMAX® W6 support started to decline. Overall, more than 50% of the enzyme was lost when assayed at and beyond said temperature.

2.2.2. Effect of Temperature on Lipase Stability

For the analysis of thermal stability (Figure 3), incubation on free and immobilized RoL was performed at 50 °C and assayed on an hourly basis up to 6 h. The relative activity of immobilized RoL appeared to decline gradually throughout 6 h of incubation. Accordingly, the half-life of free RoL was observed after 2 h of incubation, similar to the half-life of immobilized RoL on support without additives and immobilized RoL with hen egg albumin as the additive to the support. As for immobilized RoL on a support treated with sodium caseinate and CAVAMAX® W6 the half-life at 50 °C was found to be 5 and 6 h, respectively.

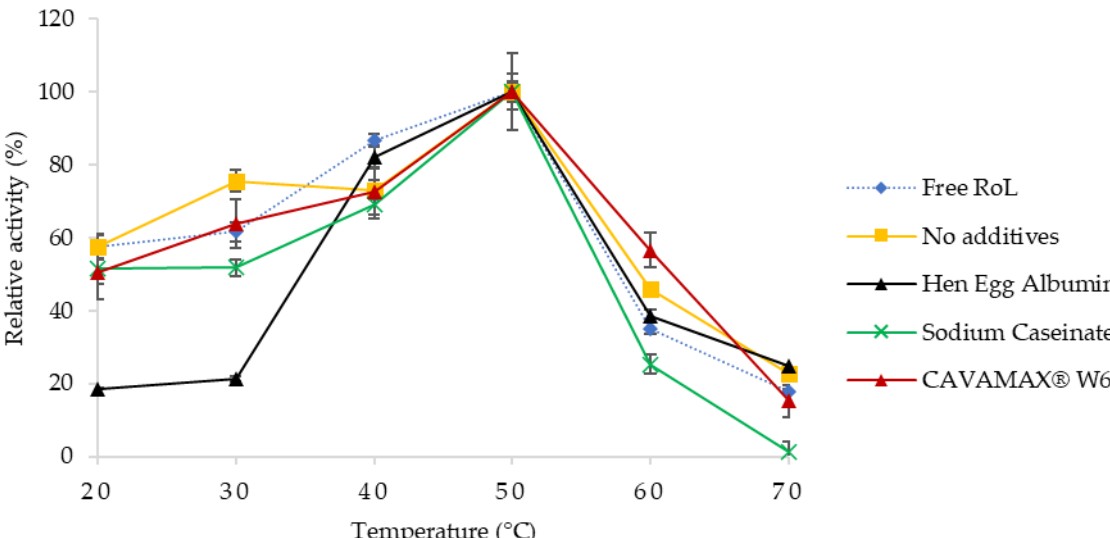

**Figure 2.** Temperature profile of free and immobilized RoL on polypropylene support with and without additives that include hen egg albumin, sodium caseinate and CAVAMAX® W6. The optimum temperature of immobilized RoL was measured at different temperatures ranging from 20 to 70 °C. All measurements were carried out in triplicate and calculated as mean ± SE (n = 3).

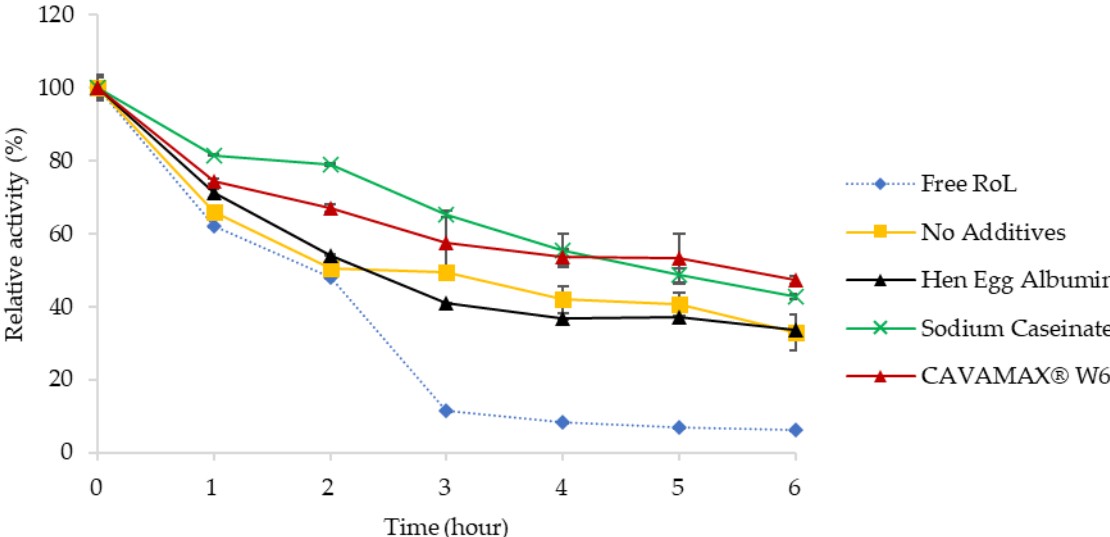

**Figure 3.** Thermal stability profile of free and immobilized RoL on polypropylene support with and without additives that include hen egg albumin, sodium caseinate and CAVAMAX® W6. The thermal stability of immobilized RoL was evaluated by incubating free and immobilized lipase at 50 °C in potassium phosphate buffer (pH 6.0) up to 6 h. The relative enzyme activity was taken at 50 °C as 100% on the non-incubated lipase. All measurements were carried out in triplicate and calculated in mean ± SE (n = 3).

The RoL immobilized on support with CAVAMAX® W6 exhibited 76% of remaining lipase activity when stored for 120 days at 4 °C (Figure 4A), while the half-life of similarly immobilized RoL was 60 days of storage at room temperature (Figure 4B). The half-life of immobilized RoL without additive at 4 °C storage was recorded at day 90, similar to RoL immobilized on support with hen egg albumin stored at 4 °C. However, the half-life of immobilized RoL without additives declined to less than 30 days when stored at room temperature. The RoL immobilized on support with sodium caseinate produced a half-life of 120 days at 4 °C and room temperature storage. After 120 days of storage, the free

form of RoL exhibited a decrease in lipase activity to 30% and 15% at 4 °C and room temperature, respectively.

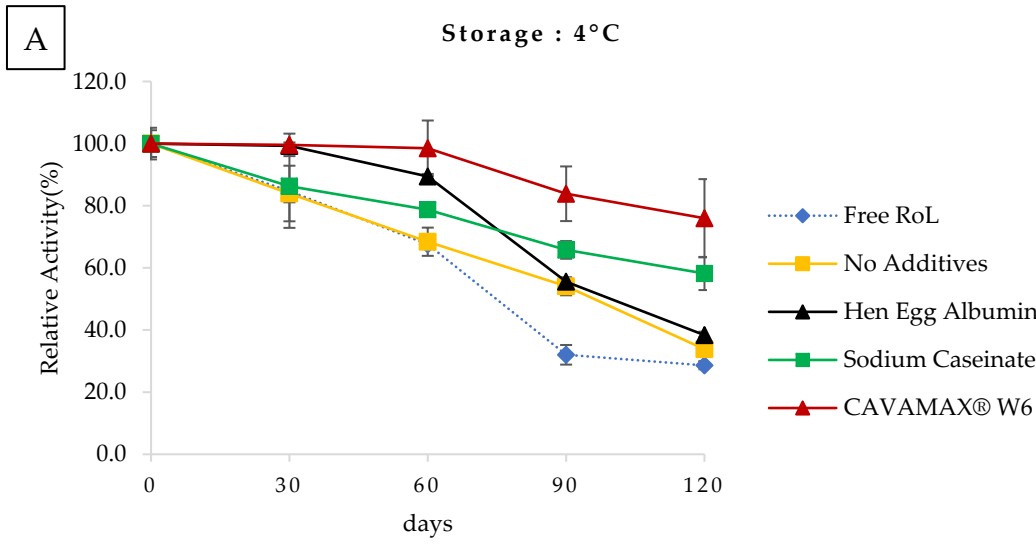

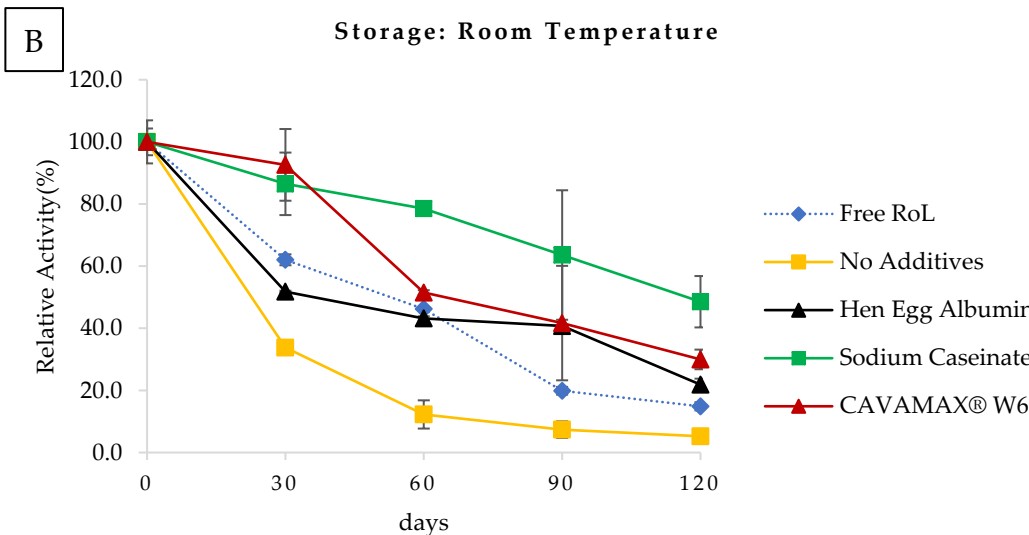

**Figure 4.** Storage stability of free and immobilized RoL on polypropylene support with and without additives that include hen egg albumin, sodium caseinate and CAVAMAX® W6. The data were a display of storage activity for 120 days in (**A**) 4 °C and (**B**) room temperature. The relative activity at 0 days of storage was referred to as 100%. All measurements were carried out in triplicate and data were calculated in mean ± SE (n = 3).

### 2.2.3. Effect of pH on Lipase Activity

The effect of pH on lipase activity of free and immobilized RoL was evaluated by an assay having a pH adjusted to a range of 4 to 9 using different types of buffer. According to Figure 5, both free and immobilized RoL were very stable at pH 4 to 6 in which phosphate buffer gave the optimum pH, which was pH 6. The highest activity was observed on RoL immobilized on support treated with sodium caseinate, which was 693.7 U/g, followed by RoL immobilized on support coated with CAVAMAX® W6 which was 465.5 U/g. In Tris-HCl buffer (pH 8), the immobilized RoL was capable of producing lipase activity higher than 50%.

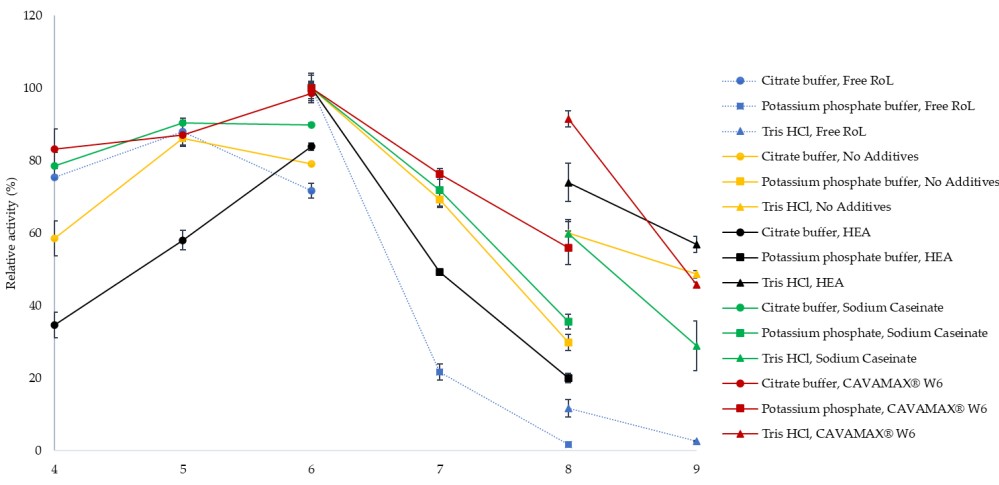

**Figure 5.** pH profile of free and immobilized RoL on polypropylene support with and without additives that include hen egg albumin, sodium caseinate and CAVAMAX® W6. Optimum pH was determined based on the activity of free and immobilized RoL at various pH comprising (●) 50 mM sodium citrate buffer (pH 4–5), (■) 50 mM potassium phosphate buffer (pH 6–8) and (▲) 50 mM Tris-HCl buffer (pH 8–9). The relative activity at different pH values was calculated by setting the optimum pH as 100%. All measurements were carried out in triplicate and data were calculated in mean ± SE (n = 3).

### 2.3. Scanning of Electron Microscopy (SEM)

SEM micrographs as disclosed in Figure 6 revealed the morphology of polypropylene supports having pores structure of nonuniform size. Accordingly, the difference in the size of the particle of the support on the reference of before and after immobilization was found to be between 1.131–1.910 mm. The morphology of the support was also reported to be similar to Silva et al. 2015. After immobilizing RoL on treated support, the area of porosity became less obvious (Figure 6G–I). This indicates that the void areas inside the pores are filled up with lipases.

### 2.4. Fourier-Transform Infrared Spectroscopy (FTIR)

FTIR was used to confirm the attachment of enzyme on the surface of the matrix, understanding the nature of bonds involved in the attachment of enzyme and conformational changes of enzyme after immobilization. The appearance of 1700–1600 cm$^{-1}$ bands indicates the presence of a peptide bond, C=O stretching vibration. A broad band observed at a range of 3000 to 3700 cm$^{-1}$ originated from stretching vibrations of O–H and amine (N–H) bonds. Moreover, the changes of spectra exhibited by the grey shades in Figure 7 indicated the presence of enzyme on the surface area of support.

### 2.5. Enzymatic Esterification of Oleic Acid

The synthesis of ethyl-oleate from oleic acid and ethanol (ratio 1:2) using free and immobilized RoL was evaluated with and without n-hexane. Compared to other organic solvents, n-hexane was selected as the main solvent for the catalytic reaction due to hydrophobicity factor that increased the stability of lipase [15]. Based on Figure 8, it was capable of achieving high conversion of ester with the use of immobilized RoL. With the presence of additive on the support such as hen egg albumin, a higher ester yield was attained of 78%. Other additives such as sodium caseinate contributed to 76% of ester while CAVAMAX® W6 contributed to 66% of ester in the enzymatic esterification with n-hexane. Unlike the immobilized RoL, free RoL failed to convert ester in organic media, which leads to low conversion of ester (36%) as compared to RoL without organic solvent that yielded ester at 59%. Such an observation on ester yields indicates the effect of organic solvent towards enzyme aggregation. To avoid lipase aggregation, immobilized RoL was utilized as the lipase was configured to be more accessible to substrate [16].

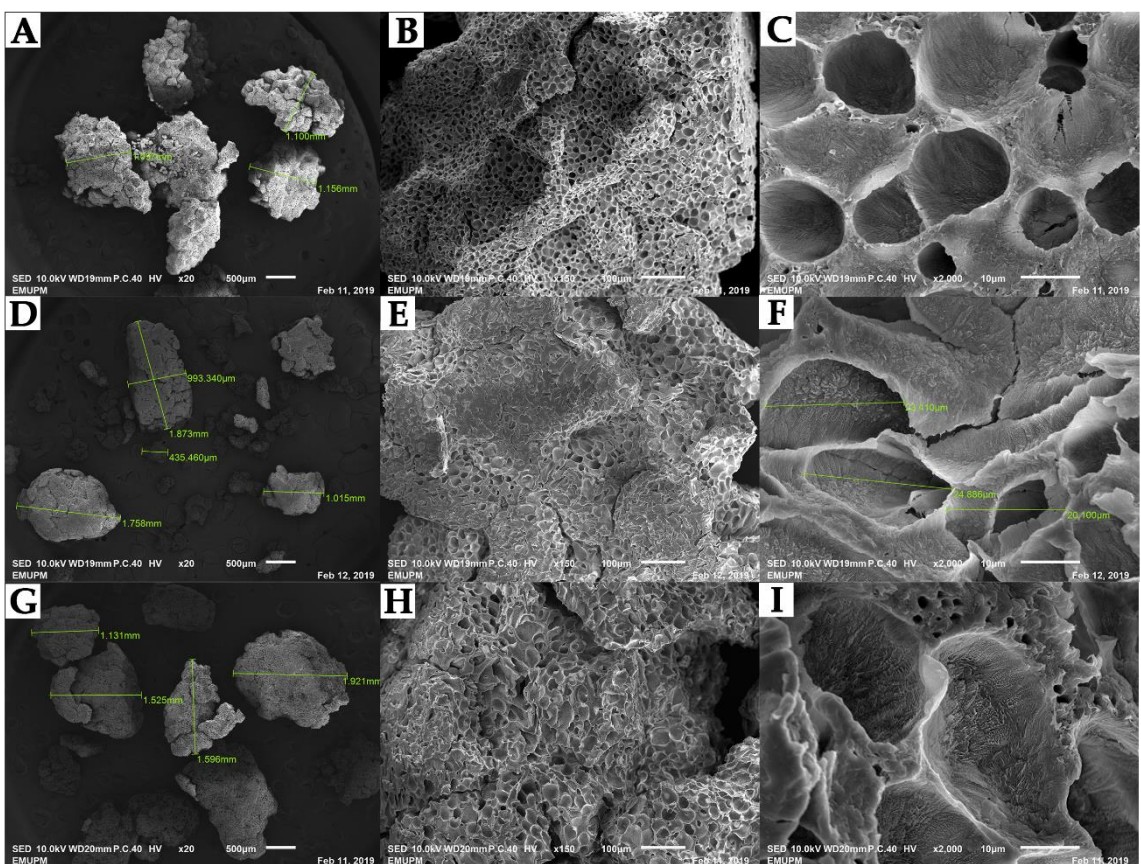

**Figure 6.** Scanning electron micrograph analysis of empty and immobilized RoL at 20×, 150× and 2000× magnification. Images (**A–C**) refer to empty polypropylene support. Images (**D–F**) refer to immobilized RoL on support without additives. Images (**G–I**) refer to immobilized RoL on support treated with hen egg albumin.

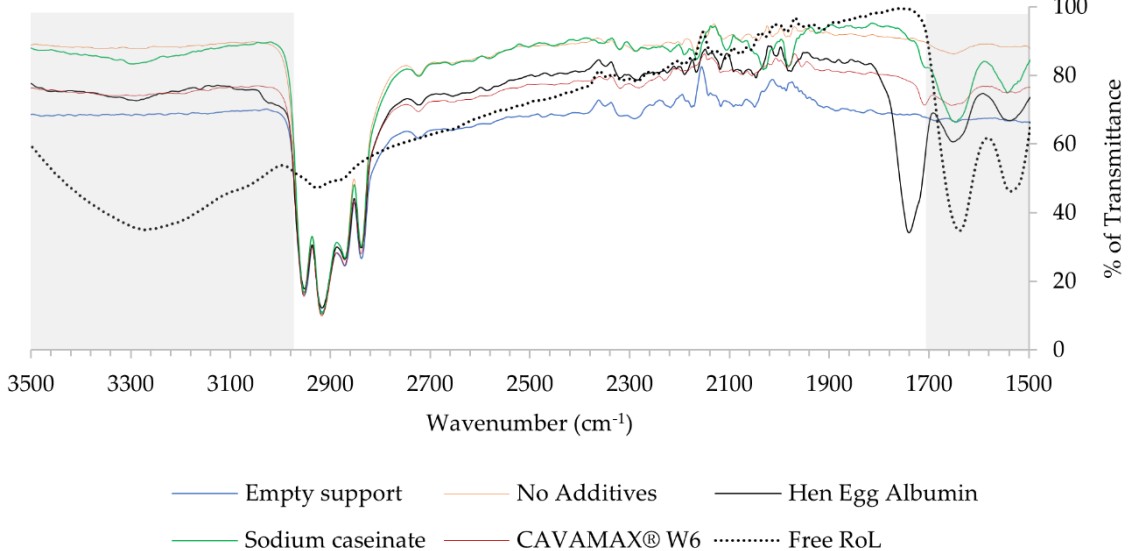

**Figure 7.** Comparison of infra-red spectra of free RoL, empty polypropylene support and immobilized RoL on polypropylene support with and without additives. The spectral changes (grey shade) harbouring bands from 3500–3000 $cm^{-1}$ indicating the N–H bending and 1700–1600 $cm^{-1}$ for the stretching of C=O.

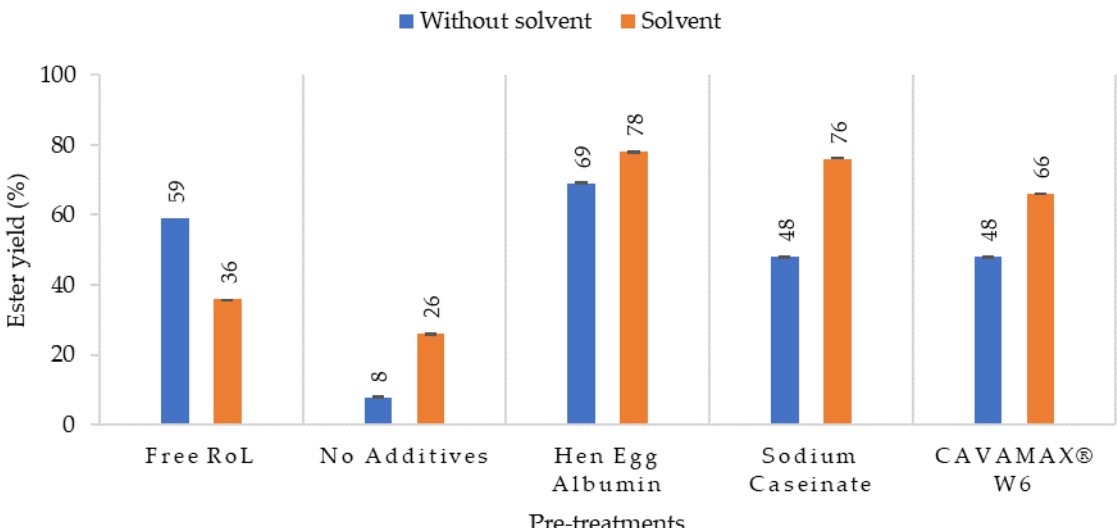

**Figure 8.** Percentage conversion of ethyl oleate catalyzed by free and immobilized RoL on polypropylene support with and without additives. The esterification was conducted in the hexane and without hexane at 50 °C with a molar ratio of oleic acid to ethanol that was 1:2.

## 3. Discussion

### 3.1. Immobilization of RoL on Polypropylene Support

Due to the nature of the polypropylene support surface that is nonionic and highly hydrophobic, a monolayer of a nonionic surfactant such as Tween 20 was applied on the surface of supports. The coating increased the membrane hydrophilicity during immobilization. Additives were used to treat the supports and improved the enzyme activity during the esterification reaction. The additives in the likes of hen egg albumin and sodium caseinate have been referred by Bosley and Peilow, 1993 for their use in the adsorption of lipase on hydrophobic support [14]. Both additives were of nonlipase proteins that contained hydrophilic and hydrophobic groups, giving good emulsifying and encapsulation properties [17]. The additives also act as a blocking agent, which minimizes the nonspecific protein binding to the surfaces [18]. The additives adsorption on the support surface was driven by the nonpolar parts of the protein up to a monolayer of the support. Besides the two additives, a polymer additive derived from food-grade alpha-cyclodextrin known as CAVAMAX® W6 (Wacker Chemie AG, Munich, German) comprises six glucose units connected by $\alpha$-1,4 linkages that were capable of forming inclusion complexes was also used in the experiment. Based on the hydrophobic nature of the cyclodextrin cavity, the adsorption and hyperactivation of lipases were executable. The effect of additives on the support for immobilization showed a significant result on improving the yield of immobilized RoL (Figure 1) above immobilization of the same without additives.

Even after immobilization, the leakage of enzyme remained as a limitation for mass transfer. In order to justify this problem, the morphology of polypropylene was examined and based on the findings; the polypropylene has a large internal surface area, showing deep structures within the pores that restrict the accessibility of lipase to the substrate. The loss of catalytic activity after immobilization could also relate to the conformational changes of lipase after immobilization or due to modification of the lipase microenvironment influenced by interactions of the support with the enzyme [11]. According to a previous study, a heterologous *Rhizopus oryzae* lipase immobilized on MP 1000 and VP OC 1600 had been prepared and used for enzyme leaching [11]. In this study, the immobilized RoL on polypropylene support with additives was configured for esterification on different systems where catalytic performance of the esterification reaction was evaluated to determine the usability of immobilized RoL.

### 3.2. Characterization of RoL Immobilized on Polypropylene Support

The effect of temperature on hydrolytic activities of free and immobilized RoL was shown in Figure 2. Different to ordinary *Rhizopus lipase,* having a stable temperature below 50 °C, the optimum temperature for immobilized RoL in the present study was 50 °C [19,20]. The main factor of a declined lipolytic activity at high temperatures was due to the disruption of hydrogen bonds that leads to weak interaction within the lipase molecules. The interaction of lipase with support was governed by hydrophobic interactions [21]. A strong hydrophobic effect on additives had resulted in the open conformation and interfacial activation of lipase during adsorption. Immobilizing RoL on support treated with CAVAMAX® W6 yielded high activity of lipase as compared to other without additives. A similar result was observed in Kolossváry et al. [18] following the high rate of hydrolysis especially of lipid hydrolysis in the presence of β-cyclodextrin, which acts to crowd the free fatty acid and accelerates lipolysis by reducing product inhibition. The half-life of free RoL was reported to be similar to the half-life of immobilized RoL without additives and immobilized RoL on hen egg albumin-treated support in which the half-life was 2 h. On the contrary, the immobilized RoL on support treated with sodium caseinate and CAVAMAX® W6 showed better stability when exposed to a prolonged incubation at 50 °C.

Duration of storage of the immobilized lipase was significantly evaluated. In the present study, the storage stability of free and immobilized RoL was compared at two temperatures that include 4 °C and room temperature. A decrease of lipase activity in free RoL was because of conformational change driven by the condition and period of storage. As for the immobilized RoL, aggregation would no longer be a problem as the enzymes were well distributed. Furthermore, rigidification of the enzyme via multiple attachments to the supports will lead to better preservation of enzyme properties under severe conditions [22].

pH also influenced the functionality of the enzyme. In this study, the pH properties of immobilized RoL remained at pH 6 in phosphate buffer. In general, immobilization can cause pH shift depending on the type of the support. When RoL was immobilized in alginate gel beads, the pH shifted from pH 8 to 7. The reason of the shifting was due to surface and residual charges on the support, for example polycationic support would result in acidic shift while polyanionic support caused in a shift to more basic pH values [23]. Therefore, immobilized enzymes are mostly active at broader pH compared to free enzymes. Figure 5 indicates that the addition of additives improves the activity and stability of RoL over a broader pH range. A similar outcome to Kharrat et al. [24] showed that the immobilization of RoL had significantly improved its stability at pH 8–8.5.

Visualizing the attachment of enzyme to the surface of support and analyzing its distribution by means of microscopy is time-consuming. However, it is imperative to determine whether the enzyme is located inside the porous structure or adsorbed on the surface of the support. To the advantage of FTIR spectra, the attachment of lipase immobilized on the support could be confirmed. Accordingly, the interactions of lipase to the support were based on the adjustment to the secondary structure of the enzyme. After immobilization, it was shown that the N–H bending, C=O stretching and N–H stretching could be of support to lipases that bind on the surface of the support [25]. Collins et al. [25] also used FTIR to analyze the secondary structure of the immobilized enzyme as part of the characterization of *Rhizomucor miehei* lipase immobilized in chitosan.

Enzymatic esterification using oleic acid and ethanol at 50 °C by shaking was conducted via two systems, in which the esterification system includes n-hexane and free n-hexane. A molecular sieve was used as a water adsorbent to prevent a reverse reaction. Similar studies on immobilization had been conducted using CaLB on a polypropylene-coated glass balls [26]. A batch lipase-catalyzed esterification of oleic acid with ethanol in solvent-free systems had also been tested [27] using immobilized *Candida* sp. [28,29]. In the present study, the production of ethyl oleate had improved with *RoL* immobilized on a hen egg albumin-treated support in the presence of solvent. A lower retention activity was obtained with *RoL* immobilized on support treated with hen egg albumin at the

oil/water interface reaction system due to the size of emulsion droplets which exceed the medium pore diameter of the support. Such an effect leads to a restricted access of lipase to substrate [11].

## 4. Materials and Methods

### 4.1. Materials

*Rhizopus oryzae* lipase (powder) was obtained from Amano Enzymes, Nagoya, Japan. Polypropylene support was purchased locally from Hynix Asia Sdn Bhd, Selangor, Malaysia. The support particle size was between 200–1500 μm, and a specific surface area was 0.78 m$^2$/g determined by the BET surface area method. The hen egg albumin was purchased from Sigma-Aldrich, Selangor, Malaysia, while sodium caseinate and CAVAMAX® W6 were locally purchased from DKSH, Selangor, Malaysia. CAVAMAX® W6 is a food-grade alpha-cyclodextrin from Wacker Chemie AG, Munich, Germany where the ring-shaped oligosaccharide is produced enzymatically from plant starch. Bertolli olive oil was purchased from a local market in Malaysia.

### 4.2. Methods

### 4.3. Immobilization of RoL in Polypropylene Support

The immobilization procedure with slight modification was carried out according to two granted patents by Bosley and Peilow, 1993 (Patent No. 5232843) and Bosley and Moore, 1998 (Patent No. USOO5773266A). Lipase solution (0.1%) was added into the treated supports mixture and was stirred for four hours. The supernatant was collected at one-hour intervals. The lipase adsorbed on the supports was filtered and recovered using Buchner funnel vacuum filtration. The wet immobilized lipase was dried using Fluid Bed Dryer TG 200 (Retsch GmbH, Germany) with drying time of 45 min at 40° and with airflow of 40 m$^3$/h. The supernatant obtained from the sampling was checked based on the hydrolytic activity left on the solution. The yield of immobilization was estimated according to the following Equation (1), where t0 is the amount of enzyme present in the buffer solution before the support was added, and t4 is the residual amount of enzyme present in the supernatant after immobilization has completed. The experiment was repeated by replacing hen egg albumin with sodium caseinate and CAVAMAX® W6. The preparation of the support and additives were skipped for experiments involving no additives.

$$\text{Immobilization yield } (\%) = \frac{(\text{t0} - \text{t4})}{\text{t0}} \times 100 \tag{1}$$

Loss of hydrolytic activity after immobilization on immobilized RoL was also observed. A total of 10 mg of immobilized RoL was incubated at 50 °C in 50 mM phosphate buffer under an agitation rate of 200 rpm for 24 h. The activity of immobilized RoL at pre- and postreaction were assayed and the amount of enzyme leached out from the support was calculated according to the Equation (2).

$$\text{Activity loss } (\%) = \frac{\begin{pmatrix}\text{Initial suspension activity-immobilized enzyme activity}-\\ \text{Supernatant activity after immobilization}\end{pmatrix}}{(\text{Initial suspension activity})} \times 100 \tag{2}$$

### 4.4. Lipase Activity Determination

Lipase activity was assayed by a colorimetry method from Kwon and Rhee, 1986 with slight modifications. About 10 mg of immobilized lipase, 2.5 mL of olive oil emulsion, and 20 μL of 20 mM CaCl$_2$ were incubated in a water bath shaker at 50 °C with agitation rate of 200 rpm for 30 min. After incubation, the enzyme reaction was terminated by adding 1 mL of 37% HCl (6 N) into the reaction mixture. A total of 5 mL of isooctane was added, and the mixture was vortexed to mix for 20–30 s each. The mixture was left to settle down for at least 30 min before separating the upper-layer and transferring the layer into a clean vial containing 1 mL of 5% (*w/v*) copper (II) pyridine. The mixture was vortexed for the second time and left to settle at room temperature until a green-colored lysate appeared at the first

layer. Using the green-colored lysate, lipase activity was measured spectrophotometrically at 715 nm and standard curve oleic acid was used to quantify said lipase activity. One unit of lipase activity was defined as the rate of 1 μmole fatty acid released per minute [30].

### 4.5. Characterization of Immobilized RoL

#### 4.5.1. Effect of Temperature on Enzyme Activity

The nonimmobilized and immobilized lipase were incubated at temperature range at 30 °C to 70 °C with 10 °C intervals for 30 min. The enzyme activity was assayed, as described at Section 4.2.

#### 4.5.2. Effect of Temperature on Enzyme Stability

The nonimmobilized and immobilized lipase samples were incubated at 50 °C for 1–5 h before the lipase assay. The residual enzyme activity was assayed, as described previously. For storage stability, the immobilized enzyme was stored at 4 °C and room temperature, and the residual enzyme activity was assayed at monthly intervals.

#### 4.5.3. Effect of pH on Enzyme Activity

The effect of pH on immobilized RoL activity was evaluated at pH 4 to 9. Lipase activity was assayed using buffers with different pH that include 50 mM sodium citrate buffer (pH 4–5), 50 mM potassium phosphate buffer (pH 6–8), and 50 mM Tris-HCl buffer (pH 8–9).

#### 4.5.4. Scanning of Electron Microscopy

The empty and immobilized lipase morphology was investigated using Scanning Electron Microscopy (SEM). Analysis was carried out using JSM-IT200 InTouchScope™ (JEOL, Japan). The supports were mounted on a specimen stub and dried overnight in the oven (40 °C) to remove the sample's moisture. Then, the sample was coated with gold before scanned by SEM. The morphologies and shapes of support were observed at various magnifications (20×, 500×, and 2000×).

#### 4.5.5. Fourier-Transform Infrared Spectroscopy

The physical characteristics of free lipase, empty supports, additives and immobilized lipase were determined by the Fourier Transform Infrared Spectroscopy Attenuated Total Reflection (FTIR-ATR) test method. Analysis was carried out at a resolution of 4 cm$^{-1}$ between 4000 and 450 cm$^{-1}$ with an average of 32 scans using the FT-IR spectrometer Nicolet 6700, Thermo Scientific, Waltham, MA, USA.

### 4.6. Enzymatic Esterification of Oleic Acid

Enzymatic esterification of oleic acid by immobilized lipase was conducted according to the modified Cea et al. 2019 method. The enzymatic esterification reaction was carried out in screw-capped bottles containing oleic acid (1.0 mmol), ethanol (2.0 mmol) and free or immobilized lipase (5 wt% based on oleic acid). The reaction was carried out with or without solvent system (1.0 mL hexane) at 50 °C, by constant shaking at 200 rpm for 18 h. The reaction was terminated by dilution with 2.0 mL of ethanol: acetone (50:50 *v/v*). The remaining free fatty acids in the reaction mixture were determined by titration with 0.1 M NaOH using phenolphthalein as an indicator. For blank determination, the reaction mixture without enzyme was used. The yield of ester, ethyl oleate was expressed as a percentage of converted oleic acid compared to the total initial fatty acid content in the reaction mixture. The conversion of ester was expressed as percentage conversion (%) according to the following Equation (3) [31–33]:

$$\text{Conversion of ester } (\%) = \frac{\text{Volume of NaOH (Control} - \text{With enzyme)}}{\text{Volume of NaOH (Control)}} \times 100 \quad (3)$$

## 5. Conclusions

This study has shown that the immobilization of RoL on support treated with additives was capable of further enhancing the activity and stability of immobilized lipase. According to the catalytic activity of immobilized RoL on support treated with CAVAMAX® W6, the enzyme was found to be more stable at 50 °C and storage stability at 4 °C. The immobilized enzyme had an optimum temperature and pH preference at 50 °C and pH 6, respectively. Overall, the immobilization of enzyme on porous support with additives had successfully enhanced the biophysical characteristics of immobilized enzyme with minimal loss in hydrolytic activity. In addition, the oleic acid esterification with RoL immobilized in support treated with hen egg albumin leads to highest ester yield compared to free-RoL and immobilized RoL on support with other additives. Therefore, the high ester yield was a proof of evidence for the efficiency of immobilized RoL in which the addition of hen egg albumin to the support was the main reason of improvement.

**Supplementary Materials:** The following are available online at https://www.mdpi.com/2073-4344/11/3/303/s1.

**Author Contributions:** F.S. conceived and designed the experiments, performed the experiments, analyzed the data, contributed reagents/materials/analysis tools, prepared figures and/or tables, authored or reviewed drafts of the paper, approved the final draft. N.F.M. conceived and designed the experiments, authored or reviewed drafts of the paper, approved the final draft. M.S.M.A. analyzed the data, approved the final draft. R.N.Z.R.A.R. conceived and designed the experiments, analyzed the data, contributed reagents/materials/analysis tools, authored or reviewed drafts of the paper, approved the final draft. All authors have read and agreed to the published version of the manuscript.

**Funding:** This research was supported by the Ministry of Science, Technology & Innovation, Malaysia (ScienceFund: 02-01-04-SF2460) and the APC was funded by Universiti Putra Malaysia.

**Data Availability Statement:** Data is contained within the article or supplementary material.

**Conflicts of Interest:** The authors declare there are no competing interests.

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
