# Peer review of "Enhanced Performance of Immobilized Rhizopus oryzae Lipase on Coated Porous Polypropylene Support with Additives"

_catalysts, doi:10.3390/catal11030303_

Round 1

Reviewer 1 Report

Comments:

Line 64-65: Not for all applications the use of organic solvents is preferable.

Clarification of equations 1 and 2. I think the terms involved in these equations need to be defined more precisely. A determination of the mass of immobilized protein per mass unit of support could be useful to help explain the results of Figure 8.

I think a further discussion would be necessary to connect the data from Table 1 and Figure 8, since immobilized lipase treated with hen egg albumin seems the most effective; however, in Table 1 it is the one with the greatest loss after immobilization.

At line 347 (conclusions) it is stated that lipase with Cavamax was the most stable. However, it should be specified what type of stability it refers to: storage stability or thermal stability.

Reviewer 2 Report

The paper deals with the problem of surface modifications of solid polymer support that is used for enzyme immobilization. I nmy view such problems were studied 40-50 years ago. There is no novelty in the paper.

The language style of the paper is poor. Some sentences almost lack any sense. The paper is unacceptable in submitted form.

The study and results presented in the paper resemble a MSc student thesis. There is a lot of results but nothing new and interesting. Many materials used in experiments are not defined as they come from local sources without any data.

I cannot recommend acceptance of the paper.

Author Response

Response to Reviewer 2 Comments

Point 1: The paper deals with the problem of surface modifications of solid polymer support that is used for enzyme immobilization. I nmy view such problems were studied 40-50 years ago. There is no novelty in the paper.

Response 1:

We thanked reviewer 2 for the comments to improve the manuscript. We agree with this. The study has emphasized the effect of the additives on the support pre-treatment for immobilization. We found a lack of study on using cyclodextrin for the support pre-treatment for lipase immobilization.  

Point 2: The language style of the paper is poor. Some sentences almost lack any sense. The paper is unacceptable in submitted form.

Response 2:

We agree with this and we have taken the initiative by submitting the manuscript for English editing for better reading and understanding.

Point 3: The study and results presented in the paper resemble a MSc student thesis. There is a lot of results but nothing new and interesting. Many materials used in experiments are not defined as they come from local sources without any data.

Response 3:

Thank you for directing this out. The study in this manuscript was originally part of the master research study. The materials already described in detail in line 294-301:

“Rhizopus oryzae lipase (powder) was obtained from Amano Enzymes, China. Polypropylene support was purchased locally from Hynix Asia Sdn Bhd, Malaysia. The support particle size was between 200 – 1500 µm, and a specific surface area was 0.78 m2/g determined by BET surface area method. The hen egg albumin was purchased from Sigma-Aldrich, Malaysia, while sodium caseinate and CAVAMAX® W6 were locally purchased from DKSH, Malaysia. CAVAMAX® W6 is a food-grade alpha-cyclodextrin from Wacker Chemie AG, where the ring-shaped oligosaccharide is produced enzymatically from plant starch. Bertolli olive oil was purchased from a local market in Malaysia.”

Point 4: I cannot recommend acceptance of the paper.

Response 4:

We thanked Reviewer 2 for the comments to improve this manuscript. We have made several changes in the manuscript, especially in the style of languages.   

Reviewer 3 Report

Authors described original and scientifically important study regarding improvement of lipase biocatalytic efficiency via immobilization on polypropylene support with additives. In my opinion it will be of interest for readers of the journal Catalysts and I suggest to accept the manuscript for publication after minor revisions taking into account suggestions and remarks described bellow.

In principle, the effect of described 2 additives was already described in the literature and the only innovation was added by utilisation of cyclodextrin additive. It is clear from achieved results, that improvements using cyclodextrin additive are replaced by deterioration of some parameters. The style of discussion regarding the later characteristic is a little dissaranged. Therefore it is necessary for clarity reasons to summarize the improvements of CAVAMAX W6 additive as compared to other additives, free enzyme and immobilized RoL for example in the form of table.

One of the most important characteristics of immobilized biocatalysts is operational stability during repeated biocatalytic cycles. What was the reason that this type of experiment is missing in this work?

Ethyl oleate as a model biocatalytic product is an important solvent for pharmacy. I suggest to mention the importance of this compound in the Introduction.

SEM imaging is rather invasive technique which may change the natural surface structure by drying and coating. Therefore any speculative conclusions regarding the origin of less shaded pores by mechanical steering (page 6, lines 149-152) should be avoided.

Page 10, Materials, lines 268-275: name of all companies, cities and countries should be provided here. I suggest also to provide more detailed specification of cyclodextrin CAVAMAX W6 in this section.

Author Response

Response to Reviewer 3 Comments

Point 1: In principle, the effect of described 2 additives was already described in the literature and the only innovation was added by utilisation of cyclodextrin additive. It is clear from achieved results, that improvements using cyclodextrin additive are replaced by deterioration of some parameters. The style of discussion regarding the later characteristic is a little dissaranged. Therefore, it is necessary for clarity reasons to summarize the improvements of CAVAMAX W6 additive as compared to other additives, free enzyme and immobilized RoL for example in the form of table.

Response 1:

We agree with this. In response to this we have added a more detailed explanation at discussion (Line 212-218):

“The additives also act as a blocking agent, which minimizes the nonspecific proteins binding to the surfaces [18]. The additives adsorption on the support surface was driven by the nonpolar parts of the protein up to a monolayer of the support. Besides the two additives, a polymer additive derived from food-grade alpha-cyclodextrin known as CAVAMAX® W6 (Wacker Chemie AG) comprises 6 glucose units connected by α-1,4 linkages that were capable to form inclusion complex was also used in the experiment. Based on the hydrophobic nature of the cyclodextrin cavity, the adsorption and hyperactivation of lipases were executable. The effect of additives on the support for immobilization showed a significant result on improving the yield of immobilized RoL (Figure 1) above immobilization of the same without additives.”

Point 2: One of the most important characteristics of immobilized biocatalysts is operational stability during repeated biocatalytic cycles. What was the reason that this type of experiment is missing in this work?

Response 2:

Thank you for pointing this out. Due to time constraints, the operational stability such as the recyclability test was not performed in this study. Hence, the data was not available to be included in this manuscript. However, we managed to perform the storage stability test for each immobilized lipase.

Point 3: Ethyl oleate as a model biocatalytic product is an important solvent for pharmacy. I suggest to mention the importance of this compound in the Introduction.

Response 3:

Thank you for the comment. We have added a new statement in Line 69 -73:

“The use of immobilized lipases in non-aqueous reaction media is highly preferred over aqueous solutions for industrial application where reverse reaction can be applied for biocatalysis. In the present study, the immobilized RoL was subjected to esterification on oleic acid to produce, ethyl oleate. Ethyl oleate is a fatty ester commonly used for fuel additive and as a solvent for pharmaceutical drug preparation.”

Point 4: SEM imaging is rather invasive technique which may change the natural surface structure by drying and coating. Therefore, any speculative conclusions regarding the origin of less shaded pores by mechanical steering (page 6, lines 149-152) should be avoided.

Response 4:

We agree with this. We have removed the said statement in the new lines at between 161-164.

Point 5: Page 10, Materials, lines 268-275: name of all companies, cities and countries should be provided here. I suggest also to provide more detailed specification of cyclodextrin CAVAMAX W6 in this section.

Response 5:

Thank you for guiding this out. We have added the detailed specifications as requested in Line 294 -301.

Rhizopus oryzae lipase (powder) was obtained from Amano Enzymes, China. Polypropylene support was purchased locally from Hynix Asia Sdn Bhd, Malaysia. The support particle size was between 200 – 1500 µm, and a specific surface area was 0.78 m2/g determined by BET surface area method. The hen egg albumin was purchased from Sigma-Aldrich, Malaysia, while sodium caseinate and CAVAMAX® W6 were locally purchased from DKSH, Malaysia. CAVAMAX® W6 is a food grade alpha-cyclodextrin from Wacker Chemie AG, where the ring-shaped oligosaccharide is produced enzymatically from plant starch. Bertolli olive oil was purchased from a local market in Malaysia.”

Round 2

Reviewer 2 Report

The authors revised their manuscript extensively. All changes, corrections and amendments improved the paper and I have no formal or factual objection or comments.

However, the main comment from my previous report does not change: There is no novelty in the paper. It is detailed and neat report on immobilization and characterization of lipase. The results and the paper resemble a technical report or MSc. Thesis not the scientific paper.

The decision upon publication of the paper is up to the Editors: when the paper is published there will be harm, if the paper is not published the same conclusion is valid.